# Reasons for Being “Zero-Dose and Under-Vaccinated” among Children Aged 12–23 Months in the Democratic Republic of the Congo

**DOI:** 10.3390/vaccines11081370

**Published:** 2023-08-16

**Authors:** Daniel Katuashi Ishoso, Eric Mafuta, M. Carolina Danovaro-Holliday, Christian Ngandu, Lisa Menning, Aimé Mwana-Wabene Cikomola, Christophe Luhata Lungayo, Jean-Crispin Mukendi, Dieudonné Mwamba, Franck-Fortune Mboussou, Deo Manirakiza, Moise Désiré Yapi, Gaga Fidele Ngabo, Richard Bahizire Riziki, Adele Daleke Lisi Aluma, Bienvenu Nguejio Tsobeng, Cedric Mwanga, John Otomba, Aimée Lulebo, Paul Lusamba, Marcellin Mengouo Nimpa

**Affiliations:** 1Immunization and Vaccines Development (IVD) Program, World Health Organization (WHO), Country Office, Kinshasa 01205, Democratic Republic of the Congo; yapimo@who.int (M.D.Y.); ngabog@who.int (G.F.N.); bahizirer@who.int (R.B.R.); nguejiob@who.int (B.N.T.); mwangac@who.int (C.M.); otombatondaepengej@who.int (J.O.); nimpamengouom@who.int (M.M.N.); 2Kinshasa School of Public Health (KSPH), University of Kinshasa, Kinshasa 01302, Democratic Republic of the Congo; eric.mafuta@unikin.ac.cd (E.M.); aimee.lulebo@ksph-lisanga.org (A.L.); paul.lusamba@unikin.ac.cd (P.L.); 3Immunization, Analytics and Insights (IAI), Department of Immunization, Vaccines and Biologicals (IVB), World Health Organization (WHO), 1211 Geneva, Switzerland; danovaroc@who.int (M.C.D.-H.); menningl@who.int (L.M.); 4National Institute of Public Health, Kinshasa 01209, Democratic Republic of the Congo; nganduchristian@ymail.com (C.N.); dk.mwamba@umontreal.ca (D.M.); 5Expanded Program of Immunization, Kinshasa 01208, Democratic Republic of the Congo; aimcik@yahoo.fr (A.M.-W.C.); christophe.luhata@pevrdcongo.cd (C.L.L.); mukendijean2@gmail.com (J.-C.M.); 6Communicable and Noncommunicable Diseases Cluster, World Health Organization Inter-Country Support Teams Central Africa, Libreville BP 820, Gabon; mboussouf@who.int; 7United Nations Children’s Fund (UNICEF) Country Office, Kinshasa 01204, Democratic Republic of the Congo; dmanirakiza@unicef.org; 8Public Health Section, Higher Institute of Medical Techniques of Nyangezi, Sud-Kivu 11213, Democratic Republic of the Congo; 9Independent Researcher, Moroni 99397, Comoros; alumaadele@yahoo.fr

**Keywords:** zero-dose, under-vaccinated, behavioral and social drivers of vaccination (BeSD), Democratic Republic of the Congo

## Abstract

(1) Introduction: The Democratic Republic of the Congo (DRC) has one of the largest cohorts of un- and under-vaccinated children worldwide. This study aimed to identify and compare the main reasons for there being zero-dose (ZD) or under-vaccinated children in the DRC. (2) Methods: This is a secondary analysis derived from a province-level vaccination coverage survey conducted between November 2021 and February 2022; this survey included questions about the reasons for not receiving one or more vaccines. A zero-dose child (ZD) was a person aged 12–23 months not having received any pentavalent vaccine (diphtheria–tetanus–pertussis–*Hemophilus influenzae* type b (Hib)–Hepatitis B) as per card or caregiver recall and an under-vaccinated child was one who had not received the third dose of the pentavalent vaccine. The proportions of the reasons for non-vaccination were first presented using the WHO-endorsed behavioral and social drivers for vaccination (BeSD) conceptual framework and then compared across the groups of ZD and under-vaccinated children using the Rao–Scott chi-square test; analyses were conducted at province and national level, and accounting for the sample approach. (3) Results: Of the 51,054 children aged 12–23 m in the survey sample, 19,676 ZD and under-vaccinated children were included in the study. For the ZD children, reasons related to people’s thinking and feelings were cited as 64.03% and those related to social reasons as 31.13%; both proportions were higher than for under-vaccinated children (44.7% and 26.2%, respectively, *p* < 0.001). Regarding intentions to vaccinate their children, 82.15% of the parents/guardians of the ZD children said they wanted their children to receive “none” of the recommended vaccines, which was significantly higher than for the under-vaccinated children. In contrast, “practical issues” were cited for 35.60% of the ZD children, compared to 55.60% for the under-vaccinated children (*p* < 0.001). The distribution of reasons varied between provinces, e.g., 12 of the 26 provinces had a proportion of reasons for the ZD children relating to practical issues that was higher than the national level. (4) Conclusions: reasons provided for non-vaccination among the ZD children in the DRC were largely related to lack of parental/guardian motivation to have their children vaccinated, while reasons among under-vaccinated children were mostly related to practical issues. These results can help inform decision-makers to direct vaccination interventions.

## 1. Introduction

Before the COVID-19 pandemic, in 2019, global routine childhood vaccination programs provided protection for 86% of children, dramatically reducing the catastrophic effects of diseases such as measles, polio, diphtheria, tetanus, pertussis and others on children. Vaccination is one of the most cost-effective interventions for well-being and global development [1].

Despite the benefits of routine immunization, vaccination coverage remains insufficient in many places. For instance, nearly 25 million children remained under-vaccinated in 2021, almost six million more than the year before the COVID-19 pandemic started. The number of “ZD children”, defined as children not receiving any dose of a diphtheria–pertussis–tetanus-containing vaccine, globally went up from 13.6 million (2019) to 18.2 million (2021). ZD children are found in many communities, including in urban slums, areas suffering from conflict, and in rural remote hard-to-reach areas [1,2]. Gavi, the Vaccine Alliance, through its strategic plan, aims to reduce by 25% the number of children who have not received any vaccine by 2025, and, as per the Immunization Agenda 2030 (IA2030), to reduce the number of ZD by 50% or more by 2030 [3].

The Democratic Republic of the Congo (DRC) is a large country in the central part of the African continent, with a population of approximately 115.7 million (2021) and an estimated 4,037,161 surviving infants according to data used by the Expanded Program on Immunization (EPI). The DRC’s health system has three levels (peripheral, intermediate and national) and routine immunization is an essential preventative intervention for health facilities [4]. In the DRC, the complete and systematic vaccination of children under one against 11 vaccine-preventable diseases (VPDs) constitutes a right for the child and a duty for the parents, the government and the national community. Vaccination is free and compulsory for all children. To this end, the Congolese State mobilizes the necessary resources with support from immunization partners. Recommended vaccines include those against tuberculosis, poliomyelitis, diphtheria, tetanus, pertussis, viral hepatitis B, meningitis, pneumonia due to pneumococcus and Hib, rotavirus diarrhea, measles and yellow fever. To improve immunization, the DRC and its partners launched an emergency plan to strengthen EPI, called “Plan Mashako”, which works to strengthen routine immunization activities to prevent VPD outbreaks by increasing immunization coverage [4]. However, the number of ZD and under-vaccinated children in the DRC remains one of the highest in Africa and in the world [1,5]. As per the household vaccination coverage survey led by the Kinshasa School of Public Health (KSPH) in 2021, around 19.1% of children between 12 and 23 months old had never been vaccinated with a DTP-containing vaccine, totaling approximatively 771,098 ZD. Around 25.5% of children aged between 12 and 23 months were under-vaccinated, totaling approximately 1,029,476 children [6]. These children are, therefore, at high risk of contracting VPDs and of causing recurrent outbreaks of poliomyelitis, measles and yellow fever [4]. 

To increase vaccine uptake, and target interventions, it is important to better understand why some children are never vaccinated or why they do not receive all their recommended vaccines. Since 2018, a multidisciplinary working group was established by the World Health Organization and UNICEF to review the tools that existed to understand reasons for non-vaccination, as well as barriers and enablers to immunization. This working group was also tasked with developing a framework. This gave birth to the behavioral and social drivers (BeSD) of vaccination framework and tools. BeSD has four domains: (1) thinking and feeling, (2) social processes, (3) motivation and (4) practical issues [7].

This study was conducted to identify and compare the reasons for non-vaccination among ZD children and under-vaccinated children in the DRC in 2021, using the BeSD framework, which in turn should enable development and implementation of tailored communication and operational interventions to increase vaccination coverage.

## 2. Materials and Methods

### 2.1. Survey Design and Analysis

This cross-sectional secondary survey analysis aimed at identifying the reasons caregivers provide for the non-vaccination of their children in the DRC in 2021 and comparing these reasons between zero-dose children and under-vaccinated children. This study was nested from a large nationwide vaccination coverage survey targeting children aged 6–23 months conducted between December 2021 and February 2022 (in 511 health zones (98.5%) in the 26 provinces of the DRC). The analysis presented here includes the sub-group of children aged 12–23 months at the time of the interview.

### 2.2. Sampling

This survey has been described elsewhere [6]. Briefly, survey sampling was conducted to establish provincial representativeness. 

In each one of the 26 provinces of the DRC, 5 health areas were selected from all accessible health zones (511 of 519); then, 30% of avenues/villages in the selected health areas were sampled, and finally, 34 of all households with 1 or more eligible child were included as part of the survey. Probabilities of selection were calculated to allow conducting of weighted analyses. 

### 2.3. Data Collection

Following piloting and exhaustive training, data were collected by teams including surveyors and supervisors, using tablets with the SurveyCTO application, through in-person interviews. Vaccination status was ascertained by the transcription of data from cards or other records available in the house, and for those without records at home, data from registers in facilities, where available, and from caregiver recall if no documentation was available. Questions about the reasons for non-vaccination were asked of mothers/caregivers of zero-dose or under-vaccinated children.

Data about households (including GPS coordinates) and individual children were transmitted from teams’ tablets to a secure server established for the survey. Data quality checks were implemented in real time by the field supervisor and the survey team. 

A total of 11,019 mothers/caregivers of zero-dose children and 10,795 of those of under-vaccinated were included in the analysis and were asked about the reasons related to vaccination status.

### 2.4. Variables Used

The outcome variables of interest in the context of this study were zero-dose and under-vaccinated children. Zero-dose children were those children aged between 12 and 23 months at the time of survey who had not received any dose of pentavalent (vaccine against diphtheria, pertussis, tetanus, B hepatitis and *Hemophilus influenzae* type b) verified using card or other document, or according to caregiver recall. Under-vaccinated children were defined as those who had not received the third dose of pentavalent vaccine, while the first dose had been received, also based on documented evidence or caregiver recall. 

Regarding the reasons for non-vaccination, the behavioral and social drivers of vaccination (BeSD) framework recommended by WHO and global partners for analysis related to immunization was used as a reference [7]. This conceptual framework separates determinants of vaccination into four categories: category 1 relates to people’s thinking and feelings, category 2 relates to the social processes, category 3 focuses on the motivation factors or willingness to be vaccinated and category 4 relates to programmatic and practical factors. Category 4 is further stratified into three groups: (1) geographical and financial barriers, (2) organizational barriers for immunization services and (3) systemic barriers beyond EPI control, which indirectly affect the EPI program. 

Variables related to category 1 (thinking and feeling) included a mother being too busy and relegating the vaccination of her child, fear of side effects, not believing in vaccination, ignoring the need for vaccination and misinformation about contraindications. Those related to category 2 (social processes) included rumors, family problems, including maternal illness, sick child, religious or ethnic censorship and, in the scenario of this survey, fear of COVID-19 vaccine. Category 3 (motivation) had the parent/guardian’s intention to obtain all recommended vaccines for their children as a variable. In category 4 (practical factors), the variables relating to geographical and financial barriers included vaccination site being too far away, high cost of vaccination (though vaccination is supposed to be free, payment is sometimes demanded [6]) or pre consultation. Those relating to organization of immunization services barriers included vaccination session canceled, vaccination schedule not known, long wait times, unknown vaccination site, inappropriate timing of vaccination and absent vaccinator. Those relating to logistic factors included vaccine not available, freedom-killing factors and barriers to movement (insecurity including war or armed, ethnic or community conflicts, COVID-19 movement restrictions and providers’ strike). The latter two factors were particular to this survey, given that it was conducted in the midst of the COVID-19 pandemic and also that it coincided with health worker strikes that started in June 2021.

### 2.5. Analyses

A descriptive analysis of the reasons for non-vaccination declared by the respondents was conducted, by province, and reported as frequencies and percentages with weighted confidence intervals. The proportions of these reasons for non-vaccination declared by the respondents were first presented according to the BeSD framework and comparisons were made between groups: zero-dose vs. under-vaccinated children using the Rao–Scott chi-square test with expected minimum above 5. Analyses were conducted using Stata version 17 (Stata Corporation, College Station, TX, USA). Sample weights were included during the estimations of rates and “svy” command was used to account for the complex sampling design.

### 2.6. Ethical Considerations

Ethical approval was sought for the vaccination coverage survey from the Kinshasa School of Public Health Ethical Committee before data collection (approval: ESP/CE/175/2021). Health and politico-administrative authorities at the local level also provided their authorization. Oral informed consent was obtained from potential study participants before starting the interview. Potential survey participants were informed about the nature of the survey including objectives, risks and benefits, the confidentiality of their replies, the contact details of study personnel and that they were free to participate or not without any negative consequence. Confidentiality was maintained and the dataset was anonymized.

## 3. Results

### 3.1. Reasons Related to People’s Perceptions and Feelings

Out of the 19,676 children aged 12–23 months who were ZD and incompletely vaccinated included in the study, the reasons related to people’s thinking and feelings that may have contributed to reducing parents’ motivation to have their children vaccinated (mother being too busy and relegating the vaccinating her child, fear of side effects, not believing in vaccination, ignoring the need for vaccination and misinformation about contraindications) were cited for more than half of zero-dose children (64.03%), significantly higher than for under-vaccinated children (44.66%), *p* < 0.001 (Table 1).

The provinces with proportions above the national level of this category for ZD children were: Maindombe (84%), Tshopo (80%), Lomami (71%), Maniema (70%), Lualaba (68%), Haut Katanga (67%), Kasai Oriental (67%), Bas Uele (66%), Kasai Central (66%), Kwango (66%) and Haut Uele (66%) (Figure 1).

### 3.2. Reasons Related to Social Processes

Reasons related to social processes (rumors, family problems, including maternal illness, sick child, religious or ethnic censorship, fear of Covid19 vaccine) were cited for nearly one-third of zero-dose children (31.13%), which was significantly higher than for under-vaccinated children (26.19%), *p* < 0.001 (Table 1).

The provinces with proportions above the national level of this category for ZD children were: Kongo Central (55%), Haut Lomami (54%), Maindombe (47%), Tanganyika (46%), Lualaba (44%), Equateur (41%), Bas Uele (39%), Mongala (38%), Kasai Central (37%), Kasai Oriental (36%), Maniema (36%), Ituri (33%), Lomami (33%) and Nord Kivu (32%) (Figure 1).

### 3.3. Intention Not to Have Child Vaccinated

Regarding people’s intention not to vaccinate their children, more than four in five parents/guardians of zero-dose children (82.15%) indicated not being in favor of their children receiving “all” of the recommended vaccines, which was significantly higher than for under-vaccinated children (68.03%), *p* < 0.001 (Table 1).

### 3.4. Programmatic and Practical Reasons

For the category “practical issues” (reasons likely to limit geographic and financial access, those related to the organization of health facilities and freedom-killing factors and movement restrictions) were cited for more than one-third of zero-dose children (35.60%), which was significantly less than for under-vaccinated children (55.60%), *p* < 0.001 (Table 1).

The provinces with proportions above the national level of this category for under-vaccinated children were: Tanganyika (86%), Sud Ubangi (78%), Nord Kivu (73%), Tshuapa (68%), Kwilu (65%), Sud Kivu (64%), Sankuru (63%), Nord Ubangi (63%), Ituru (60%), Lualaba (58%) and Kwango (57%) (Figure 2). 

Below are the results by sub-categories.

#### 3.4.1. Reasons Likely to Limit Geographic and Financial Access

Reasons likely to limit geographic and financial access were cited for nearly one-tenth of zero-dose children (9.62%), which was higher than for under-vaccinated children (7.76%), *p* < 0.001 (Table 1).

#### 3.4.2. Reasons Related to the Organization of Immunization Services Barriers

Reasons related to the organization of health facilities were cited for nearly one-third of zero-dose children (29.57%), which was significantly less than for under-vaccinated children (50.00%), *p* < 0.001 (Table 1).

#### 3.4.3. Reasons Relating to Freedom-Killing Factors and Movement Restrictions

Reasons likely to limit geographic and financial access were cited for nearly one-tenth of zero-dose children (9.62%), which was significantly higher than for under-vaccinated children (7.76%, *p* < 0.001) (Table 1).

## 4. Discussion

Out of a total of 19,676 children aged 12–23 months who were ZD and under-vaccinated included in the study, the reasons related to people’s thinking and feelings as well as social processes that may reduce parents’ motivation to have their children vaccinated were significantly more cited for ZD children. Notably, more than three-quarters of parents/guardians of these children referred to not being in favor of their children receiving “all” of the recommended vaccines. Programmatic and practical reasons were significantly more cited for under-vaccinated children. There was important variation between provinces, with some provinces ranking higher for multiple barriers to vaccination being cited.

The reasons that may reduce parents’ motivation to have their children vaccinated included: “mother too busy”, fear of side effects, not believing in vaccination, ignoring the need for vaccination, misinformation about contraindications, rumors, family problems, including maternal illness, sick child, religious or ethnic censorship and fear of the COVID-19 vaccine. 

In relation to busy mothers, it is known that mothers may not have time to bring the child to a vaccination site because often the vaccination sessions take place at the same time as other activities that mothers have to prioritize. A busy mother as a reason for non-vaccination has been cited in several other studies [8,9,10,11,12,13,14,15,16,17,18,19].

The fear of side effects, not believing in vaccination, ignoring the need for vaccination and misinformation about contraindications naturally results in higher parental reluctance to have their children vaccinated. Several studies have found a link between a mother’s feelings about vaccines and a child’s vaccination status [20,21,22,23]. In these studies, mothers who had negative feelings about vaccines—linked to the beliefs listed earlier—were less likely to vaccinate their children. 

Rumors, religious or ethnic censorship, fear of the COVID-19 vaccine and the circulation of false information through the use of family or religious networks regarding vaccines certainly fuel mistrust. For example, beliefs that vaccines are made up of anti-fertility substances that can destroy female eggs and damage the reproductive system have been cited in several studies [24,25,26]. It should be noted that traditional and religious leaders can have an important role because as opinion leaders and guardians of traditions, they are generally highly respected, considered, accepted and listened to by their respective communities [27].

As far as family problems are concerned, the vaccination of children can be relegated to the background when faced with illness affecting mother or child. This pushes people to seek curative care to solve a current problem, forgetting or relegating preventive programs.

Downstream in the motivation chain of parents to have their children vaccinated, programmatic and practical reasons were significantly more cited for under-vaccinated children compared to ZD children. These reasons include geographic and financial access (vaccination site too far and high cost of vaccination or pre consultation), the organization of health facilities (vaccination session canceled, vaccination schedule not known, long wait, unknown vaccination site, inappropriate timing of vaccination, absent vaccinator and vaccine not available), freedom-killing factors and civil or armed movement restrictions (insecurity including war or armed, ethnic or community conflicts, COVID-19 restrictions and providers’ strike).

Concerning the vaccination site being too far away, the explanation would be based on the physical efforts that the parents would deploy and the time taken to have their children vaccinated, especially if they have to travel on foot. Studies conducted in particular in Kenya, Mozambique and Ethiopia, including a systematic review in sub-Saharan Africa, have found a link between the distance a family has to travel to the vaccination site and the vaccination status of the child [28,29,30,31,32]. Families who would have to walk for more than an hour to reach the vaccination site were up to 18 times less likely to vaccinate their child compared to those who were nearby. 

Several studies supporting the free vaccine policy have shown that it plays a crucial role in improving immunization coverage [33,34]. The fact that having to pay for services is highlighted by this study flags a practice that is not in line with the DRC’s free vaccination initiative, which aims at avoiding financial barriers and giving children easy access to immunization. 

Negative previous experiences when taking a child to be vaccinated may inform the perceptions of how problematic these practical barriers are: canceled vaccination session, the unknown vaccination schedule, the long waiting times, the unknown vaccination site, the unsuitable time of vaccination, the vaccinator absent or the vaccine not available. Hence, the non-vaccination of children can affect even families who were motivated to obtain all vaccines. These children would have probably been fully vaccinated if health facilities had facilitated, instead of blocking their access to vaccines. Several other studies carried out in developing countries have also found the same reasons linked to the organization of health structures to be at the root of the non-vaccination of children [10,12,13,16,17,18,35,36,37,38,39,40,41,42,43,44,45]. There are many opportunities for families to be exposed to vaccination messages, such as: curative consultations, vaccination campaigns, prenatal consultations that reach up to 82.4% of the national target according to the results of the 2018 DRC Multiple Indicator Cluster Survey [35]. A question that arises, among many others, relates to the content of the exchanges between health or community workers and families during the various contacts. Hence, the need to strengthen the competencies of clinical or community workers to convey vaccination messages, including practical information to facilitate effective access. 

With regard to reasons related to insecurity, war, armed, ethnic or community conflicts, COVID-19 restrictions or even the providers’ strike, the non-vaccination of children would be linked to the difficulties encountered by the vaccination teams in accessing these areas to provide the vaccines intervention to the children, and especially that the means of transportation usually used for the vaccines and the teams become unusable. Many studies have shown that children were less vaccinated in the most difficult to access areas, particularly in areas affected by armed conflict [29,46]. 

### Limitations

This study has several limitations. It reflects a cross sectional data collection from one time-point, and it does not provide longitudinal data collection with subsequently high risk for selection biases. Communities not included in the sampling frame may have been left out, and such communities may also have their own reasons for non-vaccination. However, participation was high, only seven health zones (<2%) were excluded due to insecurity and non-response was only 0.3% (86,920 of 87,166 selected households participating). Misclassification of the outcome of interest is possible, as the survey relied on cards and recall when a card was not available, and card availability varied significantly between the different provinces [6]. The survey was implemented in the midst of the COVID-19 pandemic, and this may have affected the way that different factors played a role as barriers to immunization. Information and desirability biases cannot be excluded either. We also did not compare the reasons indicated by parents/caregivers of zero-dose and under-vaccinated children with perceptions from those with children who were fully vaccinated. Nevertheless, we believe that the differences between the proportions of factors related to the different BeSD categories between ZD children and under-vaccinated ones are not only statistically significant (large sample size), but also meaningful, and merit further exploration.

## 5. Conclusions

This study found that in the DRC zero-dose children were not being vaccinated in large and meaningful numbers and that this was linked to challenges related to the motivation of parents or guardians towards vaccination, while for children who were under-vaccinated, the reasons they did not complete their immunization schedule were more often linked to health system programmatic and practical issues. Also, there were important provincial variations in the reasons given by parents/caregivers of the ZD and the under-vaccinated children. Thus, these results can serve to guide further and more localized studies to ultimately help decision-makers implement effective interventions to improve vaccination coverage in the DRC and reach the goal of not leaving anyone behind.

## Figures and Tables

**Figure 1 vaccines-11-01370-f001:**
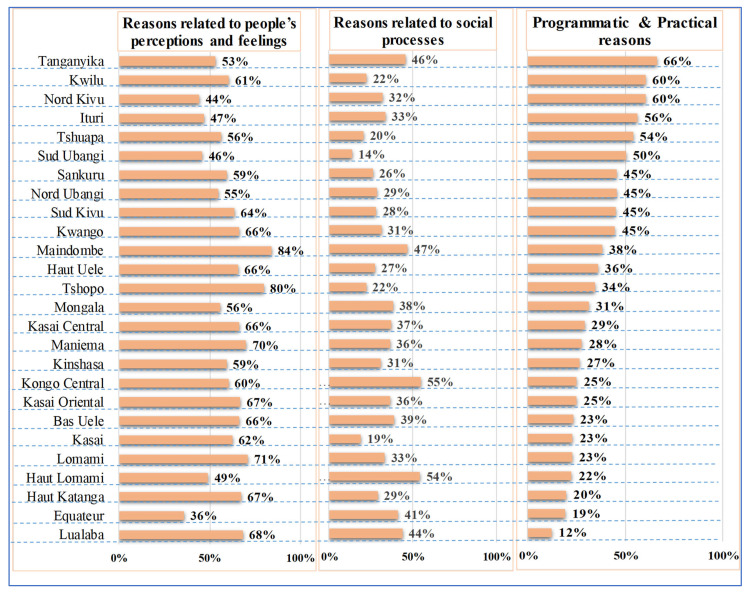
Reasons for zero-dose children by BeSD category (people’s perceptions and feelings, social processes and practical issues) and by province, DRC, 2021.

**Figure 2 vaccines-11-01370-f002:**
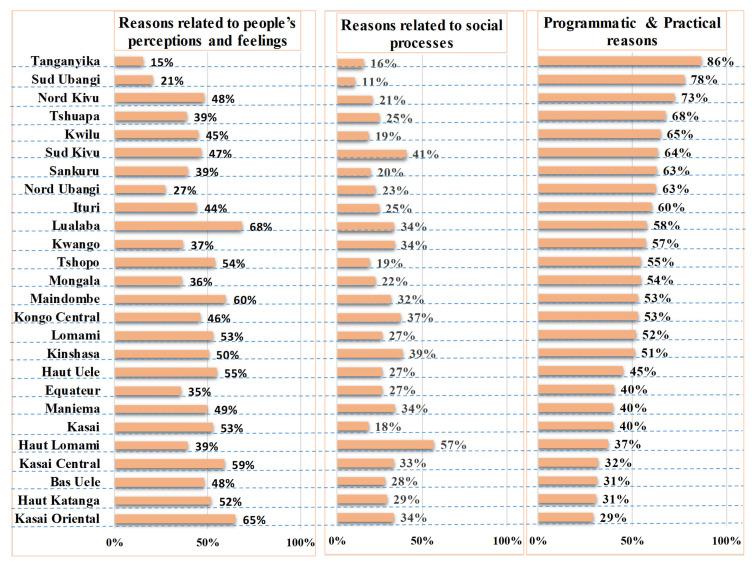
Reasons for under-vaccination of children by BeSD category (people’s perceptions and feelings, social processes and practical issues) and by province, DRC, 2021.

**Table 1 vaccines-11-01370-t001:** Reasons for non-vaccination and comparison between zero-dose children and under-vaccinated, DRC, 2021.

Variables		Zero-Dose Children (*n* = 10,765) Weighted % (95% CI)	Under-Vaccinated Children (*n* = 8911) Weighted % (95% CI)	*p*
Reasons related to people’s perceptions and feelings “Confidence in vaccine benefits”		64.03 (63.91–64.14)	44.66 (44.53–44.79)	<0.001
	Mother too busy (vaccination relegated)	30.53 (30.43–30.65)	34.42 (34.30–34.54)	
	Fear of side effects	19.22 (19.13–19.32)	6.47 (6.41–6.53)	
	Not believing in vaccination	18.78 (18.69–18.90)	2.40 (2.36–2.44)	
	Ignoring the need for vaccination	18.08 (17.99–18.20)	7.95 (7.89–8.03)	
	Misinformation about contraindications	3.20 (3.16–3.24)	1.20 (1.17–1.22)	
Reasons related to social processes “Family norms”		31.13 (31.02–31.25)	26.19 (26.07–26.30)	<0.001
	Rumors	12.08 (12.00–12.16)	5.30 (5.25–5.36)	
	Family problems, including maternal illness	10.64 (10.56–10.71)	11.42 (11.34–11.50)	
	Sick child	8.68 (8.61–8.88)	12.32 (12.24–12.40)	
	Religious or ethnic censorship	3.85 (3.80–3.90)	0.23 (0.22–0.25)	
	Fear of COVID-19 vaccine	0.30 (0.28–0.31)	0.22 (0.21–0.23)	
Intention not to have child vaccinated Parents/caregivers who do not want their child to receive “all” recommended vaccines		82.15 (82.05–82.23)	68.03 (67.92–68.14)	<0.001
Practical issues		35.60 (35.49–35.72)	55.60 (55.47–55.72)	<0.001
Factors limiting geographic and financial accessibility		9.62 (9.54–9.69)	7.76 (7.69–7.83)	<0.001
	Vaccination site too far	8.39 (8.33–8.46)	6.72 (6.66–6.78)	
	High cost of vaccination or pre consultation	1.32 (1.29–1.35)	1.14 (1.11–1.17)	
Organizational factors of the health facility		29.57 (29.46–29.68)	50.00 (49.87–50.13)	<0.001
	Vaccination schedule not known	14.64 (14.55–14.72)	17.42 (17.32–17.52)	
	Vaccines not available	10.10 (10.02–10.17)	29.13 (29.01–29.24)	
	Vaccination session canceled/absent vaccinator	7.30 (7.24–7.36)	13.60 (13.51–13.69)	
	Inappropriate timing of vaccination	4.85 (4.80–4.90)	5.70 (5.64–5.76)	
	Unknown vaccination site	2.08 (2.04–2.11)	1.57 (1.54–1.60)	
	Long waiting time	1.43 (1.40–1.46)	2.46 (2.42–2.50)	
Freedom-killing factors and movement restrictions		1.22 (1.19–1.25)	1.90 (1.87–1.94)	<0.001
	Insecurity/war/armed, ethnic, community conflicts	1.22 (1.19–1.24)	1.90 (0.87–1.94)	
	COVID-19 restrictions	0.60 (0.58–0.62)	0.27 (0.25–0.28)	
	Providers’ strike	1.23 (1.21–1.26)	4.58 (4.53–4.63)	

PC = pre consultation; *n* = number of subjects; 95% IC = 95% confidence interval.

## Data Availability

The data presented in this study are available on request from the WHO-DRC office at the email address “nimpamengouom@who.int”. The data are not publicly available due to the sensitivity of certain information from health facilities.

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
