# Peer review of "Reasons for Being “Zero-Dose and Under-Vaccinated” among Children Aged 12–23 Months in the Democratic Republic of the Congo"

_vaccines, 2023, doi:10.3390/vaccines11081370_

Round 1
Reviewer 1 Report
The results obtained provide important information for increasing immunization coverage in the Republic of Congo. It is desirable that in the next study, the authors provide information on specific steps in this direction that can be taken (or are already being taken) based on their results.
Author Response
Thank you
Reviewer 2 Report
The aim of this study was to identify and compare the main causes of zero-dose (ZD) and undervaccination (UV) children in the Democratic Republic of the Congo, a country that has a high proportion of children who have never been vaccinated. The study is based on a large vaccination coverage survey conducted in all 26 provinces across the country between December 2021 and February 2022.
The article clearly defines the purpose of the study, the definitions of UV and ZD, describes outcome variables, and follows the WHO -Immunization behavioral and social-drivers-conceptual framework. A cross-sectional survey based on home interviews included almost 20k households. Reasons for ZD or UV are grouped into 4 categories: thinking/feeling, social and motivational, and practical. The main conclusion is that ZD children in the DRC are mainly due to a lack of the parents/guardian’s motivation, while incompletely vaccinated children are mainly due to practical problems.
The following are some of the positive aspects of the study/article:
· The study addresses an extremely important topic.
· The design, methods, variables, data processing, and analysis are well described.
· The sampling is described. A large sample of households is included.
· The findings/conclusions are pertinent to the guidance of public health actions.
· A significant finding was that people's thinking and feelings that may contribute to decreasing the motivation of parents to have their children vaccinated were mentioned for more than 64% of the ZD and 44.7% of the incompletely vaccinated children.
· The authors acknowledge the potential limitations associated with the single point cross-sectional design, as well as the exclusion of some geographical areas due to safety concerns, and the fact that information was self-reported.
· Helpful public health recommendations that can be used to improve the vaccination process are given especially to the EPI program, to strengthen the health system’s operations, health education, and research to assess the effectiveness of those interventions.
Author Response
Thank you
Reviewer 3 Report
This is an interesting study on a relevant topic. I just have some comments and queries for the authors, below:
- Line 104: Was the interview conducted in person?
- Line 117: Please italicize bacterial names
- Line 118: Please rephrase “even the”
- Lines 131, 236: Please rephrase “bad idea” – are you referring to “incorrect”?
- Have the rights been obtained from the initial author for republication of Figure 1?
- Table 1: It is not enough to report only the p value. The full statistical results should be presented according to statistical reporting guidelines.
- Line 329: “Approximately” is not an appropriate term here, please revise.
- This study was performed some time during the first COVID-19 waves. This should be discussed in order to provide some context regarding the waves of COVID-19 in the country at that time, and the restrictions that were in place when or just before the survey was performed. This is important to understand in order to see whether the results are likely to be limited or reflect constraints temporarily induced by containment measures due to COVID-19, or are a de facto reality expected to be present to this day in the country.
Minor spelling changes should be performed.
Author Response
- Line 104: Was the interview conducted in person?
- R) Yes, it was. This was clarified in the text
- Line 117: Please italicize bacterial names
- R) This was corrected in the manuscript and done so for Haemophilus influenzae, as this is the only bacterium genus and species mentioned, as opposed to the diseases listed not in italics as per the CDC nomenclature guidance https://wwwnc.cdc.gov/eid/page/scientific-nomenclature .
- Line 118: Please rephrase “even the”
- R) This was corrected in the manuscript
- Lines 131, 236: Please rephrase “bad idea” – are you referring to “incorrect”?
- R) This was reworded in the manuscript
- Have the rights been obtained from the initial author for republication of Figure 1?
- R) Figure 1 is removed because the rights to the original author for its republication have not been formally obtained
- Table 1: It is not enough to report only the p value. The full statistical results should be presented according to statistical reporting guidelines.
- R) Thank you for this comment. Our analysis was fairly descriptive, and as such we chose to present a simple table showing the proportions of reasons (coded in the questionnaire) reported by the respondents, and stratified according to the Behavioral and Social Drivers for vaccination (BeSD) framework criteria, for ZD children alongside those who were under-vaccinated. As we only compared proportions in the two groups using the chi-square test (as indicated in the methodology), we chose not to include the chi-square test value, as it would have not added much to the main message and crowded the table..
- Line 329: “Approximately” is not an appropriate term here, please revise.
- R) This was corrected in the manuscript
- This study was performed some time during the first COVID-19 waves. This should be discussed in order to provide some context regarding the waves of COVID-19 in the country at that time, and the restrictions that were in place when or just before the survey was performed. This is important to understand in order to see whether the results are likely to be limited or reflect constraints temporarily induced by containment measures due to COVID-19, or are a de facto reality expected to be present to this day in the country.
- R) Indeed, this study was carried out in the midst of the COVID-19 pandemic. This is one of the reasons why we explored if the restrictions of movement/confinement and also fears of Covid-19 vaccines were reasons cited by caregivers for not getting their children vaccinated. We added these reasons within the BeSD framework. Children included in the survey were born roughly between July 2020 and August 2021 (youngest children being 6 months old; data not presented). The first Covid-19 case in DRC was identified on 10 March 2020. The first wave peaked around May-June 2020. The second wave, that overwhelmed hospitals in some areas started in November 2020 and lasted through the first two months of 2021; curfew and other restrictions of movement were imposed. A third and steadier wave started in May 2021. First vaccines arrived in April 2021, but initial availability and uptake were low.[1] Another event of importance, were strikes of health workers that started in June 2021 and were of varied duration and intensity throughout the country.[2] We added text throughout the manuscript to make the Covid-19 context more apparent.
[1] https://www.exemplars.health/emerging-topics/epidemic-preparedness-and-response/essential-health-services/democratic-republic-of-the-congo?requestedPdfUrl=%2Fapi%2FPdfHandler%2FDownloadPdf%3Fid%3D%7B36A78418-47AF-48BD-B1C6-ADD4C3E8EFA1%7D%26lang%3Den&requestedPdfTitle=ECREssentialHealthandServicesDRC+Narrativev20+1
[2] https://globalpressjournal.com/africa/democratic-republic-of-congo/nurses-tipping-point-battle-regular-pay/
Reviewer 4 Report
In the manuscript submitted by Ishoso et al, the authors attempt to study the reasons for the lack of vaccination (either zero-dose or undervaccination) in the Democratic Republic of the Congo (DRC), where infant vaccination is relatively poor. The authors come to the conclusion that the predicament of unvaccinated (i.e. zero dose) children are primarily related to attitudes about vaccination, while those of under-vaccinated children are primarily related to a lack of practical access to all three doses of pentavalent vaccine.
The Reviewer believes that the paper could be accepted for publication, provided that the following concerns are addressed:
Major Issues:
- there are considerable grammatical errors throughout the manuscript. While the Reviewer has corrected some of them, the Authors should have the manuscript corrected by a native speaker;
-the authors should discuss the current policy of the DRC as it pertains to vaccination (timeline, cost, is it mandatory?)
-how were the responses weighted?
-Table I-- P values should be indicated (not just less than 0.001). For some of these values, the values between zero-dose and under-vaccinated children seems to be quite close for P<0.001
Minor Issues:
Line 54: 25 million children are unvaccinated
Line 56-57: use quotation marks
Line 63: ...of the African continent
Line 81-82: Please re-phrase sentence
Line 88: "...in the all country" should be "in the entire country"
Line 92: "..was used"
Line 118: "...or even the second dose..."
Line 166: is incompletely vaccinated the same as under-vaccinated? if so, the authors should chose one or the other, and keep it consistent throughout the manuscript
There are a number of grammatical errors. The manuscript should be edited by a native English speaker
Author Response
Major Issues:
- there are considerable grammatical errors throughout the manuscript. While the Reviewer has corrected some of them, the Authors should have the manuscript corrected by a native speaker;
R) We have had this manuscript significantly edited;
-the authors should discuss the current policy of the DRC as it pertains to vaccination (timeline, cost, is it mandatory?)
R) The current policy of the DRC with regard to vaccination was added in the introduction section.
-how were the responses weighted?
R) The weight of each of the statistical units in the sample had been taken into account during the survey and presented in the database. And in the analysis phase, sample weights were included during the estimations of proportions and “svy” command was used to account for the complex sampling design. This was made more explicit in the text.
-Table I-- P values should be indicated (not just less than 0.001). For some of these values, the values between zero-dose and under-vaccinated children seems to be quite close for P<0.001
R) All p-values are less than 1 in 1000, which is certainly related to the large sample size. The p-value presentation used was based on the template seen in other publications, including those from the journal Vaccines. We added a comment in the discussion section about the magnitude of effects, in addition to noting that the large sample size affects statistical significance.
Minor Issues:
- Line 54: 25 million children are unvaccinated
- Line 56-57: use quotation marks
- Line 63: ...of the African continent
- Line 81-82: Please re-phrase sentence
- Line 88: "...in the all country" should be "in the entire country"
- Line 92: "..was used"
- Line 118: "...or even the second dose..."
- Line 166: is incompletely vaccinated the same as under-vaccinated? if so, the authors should chose one or the other, and keep it consistent throughout the manuscript
Thank you for those corrections, they were addressed in the revised version.
Reviewer 5 Report
interesting study but what about sample size? I didn't understand how questionnaire was structured. Parts of limitations should be added to matherials and methods, even non -responders rate.
Author Response
interesting study but what about sample size? I didn't understand how questionnaire was structured. Parts of limitations should be added to matherials and methods, even non -responders rate.
R) We added more information in the manuscript to make the aspects highlighted by the reviewer clearer.